# Development and Validation of a Questionnaire for Assessing Drug Use Motives in the General Population in South Korea

**DOI:** 10.3390/healthcare12010086

**Published:** 2023-12-29

**Authors:** Joon-Yong Yang, Minhye Kim, Aeree Sohn

**Affiliations:** 1Division of Social Welfare and Health Administration, Wonkwang University, Iksan 54538, Republic of Korea; lainfox@snu.ac.kr; 2Department of Sociology, College of Social Science, Changwon National University, Changwon-si 51140, Republic of Korea; minhyekim@changwon.ac.kr; 3Department of Public Health, Sahmyook University, Seoul 01795, Republic of Korea

**Keywords:** substance use, motives, instrument, narcotics, psychotropic drugs, cannabis

## Abstract

Drug use is increasing in South Korea, necessitating evidence-based policy interventions. However, there is a dearth of reliable tools for identifying the personal and psychological factors that drive drug use among Korean adults. In this study, we developed and validated an instrument based on a motivational model to measure drug use motivation in the Korean context. A survey was administered online to 250 Korean adults aged 19–59. Of the initial 37 potential items evaluated, 23 were retained after exploratory factor analysis. Based on their relevance and factor loadings, the final scale consisted of 15 items distributed across five distinct factors: enhancement, coping, social, positive expectancy, and negative expectancy. Confirmatory factor analysis validated the strong fit of the five-factor model. Criterion-related validity was demonstrated by the significant correlations between the five factors and the selected criterion variables. This instrument can be used in research initiatives related to drug addiction and can provide basic data for policy interventions intended to curb drug addiction problems in South Korea. The novelty and broad applicability of this instrument make it invaluable for exploring the psychological underpinnings of drug use in South Korea.

## 1. Introduction

In Korea, “drugs or narcotics” usually refer to narcotics, psychotropic drugs, and cannabis. These drugs are strictly regulated by Korean law, particularly the “Psychotropic Drugs Control Act”, which defines a drug as a substance that acts on the central nervous system and is recognized as extremely dangerous and harmful to the human body if misused or abused. Korea has strict drug laws, and the possession, use, or distribution of these substances without proper authorization can have severe legal consequences [1]. In Korea, drug-related arrests increased by 68.57% over a four-year period, rising from 12,613 in 2018 to 16,044 in 2019, peaking at 18,395 in 2022 [2]. Criminological research on drug use in Korea suggests that the actual number of drug users is approximately 28 times higher than the number of detected cases [3,4]. This extrapolation places the number of drug users in the country at approximately 515,000, or 1.0% of the total population. Notably, individuals in their twenties and thirties represent a significant proportion of those arrested for drug use, with arrests among this demographic increasing from 40.6% in 2018 to 57.1% in 2022 [2]. A breakdown of arrests in 2022 showed that 12,035 people (65.43% of the total) were arrested for the illicit use of psychoactive drugs. This was followed by 3809 arrests (20.71% of the total) for cannabis use and 2551 (13.87% of the total) for the use of other narcotics and psychotropic substances.

Although the percentage of individuals consuming drugs in Korea is relatively low compared to the percentages in Western and Latin American countries, there has been an increase in these rates, highlighting the need for prompt intervention. One study calculated the economic impact of drug use on Korean society and estimated a significant cost of KRW 4.87 trillion in 2016 [4]. The multifaceted challenges of drug addiction extend beyond the afflicted individual to the immediate social environment. It is crucial to understand drug use dynamics in Korea, as evidence indicates that drug addiction prevention programs could result in potential savings of USD 18 in ancillary societal costs for every USD 1 allocated. Thus, prioritizing prevention could be a viable means of reducing societal expenditure [5].

According to motivational models, drug users typically have specific outcome expectations related to substance use [6,7]. This contrasts with the premises proposed by behavioral psychologists. Motivational theorists assert that human behavior is driven by distinct intentions and motivations, as indicated in Maslow’s hierarchy of needs [8,9,10]. These motivational paradigms categorize motives into two overarching categories: self-focused and social-focused [11,12]. While self-centered motives relate to individual responses or personal stimulus enhancement, social-centered motives revolve around interpersonal dynamics and relationships.

The genesis of the motivational model of drug use can be traced to the refinement and expansion of the Drinking Motives Measure (DMM), which postulates four key motives: recreational, coping, social, and conformity [11]. In its adaptation to cannabis use, an “expansion” motive was introduced to reflect users’ curiosity and pursuit of novel experiences [6]. This quintet of motivational typologies has been rigorously validated for both validity and reliability in the context of various drugs, particularly cannabis and opiates [13,14,15]. In a variation of the model, Jones et al. [16] integrated “pain relief” into the original motives to formulate a four-factor model. A notable effort to improve the relevance of the DMM for broader substance abuse resulted in the Substance Use Motives Measure (SUMM) [12]. The SUMM structures motives into eight domains: enhancement, social, conformity, anxiety coping, depression coping, boredom coping, self-expansion, and achievement. Additionally, some studies have examined the types of motivation in greater detail under the theoretical framework of the DMM [17,18,19,20].

Research exploring motivations for drug use in the Korean population is limited. Yu [21] examined the reasons for drug consumption in a sample of 1828 Korean adolescents and found that the prevailing motivations were seeking euphoria, social integration, relief from pain and sadness, the desire to conform to a trendy public image, yearning for adulthood, societal rebellion, combating lethargy, and weight management. However, it is important to note that Yu motivated the participants to respond. Yoo [22] examined the motivational frameworks of 136 Korean women enrolled in a conditional deferred prosecution program following drug-related offenses. Their motivations for substance use included curiosity, stress relief, social interaction, and recreational activities. The results of Yoo’s study align with those of Simons et al.’s [6] five-factor model.

In this study, we aimed to develop an assessment tool to identify the factors that lead to substance use in the Korean population. This tool focused on cannabis, narcotics, and psychotropic drugs. In this study, we recognized the importance of substance type in influencing motivational structures [12]. Instead of focusing on individuals with an addiction, the aim was to create a tool suited for application to individuals who either lacked experience with drugs or were in the early stages of experimentation. This approach is based on a practical evaluation that recognizes the increased influence of behavioral factors during the early stages of substance use compared to the addiction stage, which is inherently shaped by the physicochemical characteristics of the substance that causes an addiction.

## 2. Materials and Methods

### 2.1. Participants and Procedures

In this study, we aimed to develop a standardized instrument for adults aged 19–59. Stratified sampling was used to select 250 participants, with proportions representative of both sexes and ages across the country. A questionnaire survey was administered online through M Research Company, which has the largest research panel in Korea.

### 2.2. Survey Item Development and Selection Process

To develop the questionnaire, the research team initiated the study in early April 2023 through a comprehensive literature review. In May, references relevant to the identified theories were gathered to inform the development of the research tools. By the end of May, a draft of the survey instrument comprising 55 items related to motivations for drug use was created. In early June, expert translation and back-translation were performed on these items, and detailed consultations were held regarding the content of the questions. The expert committee consulted for this research comprised professionals from diverse fields, including healthcare, pharmacy, and various disciplines in sociology and Korean linguistics. During this process, 18 items were deemed inappropriate and thus excluded, resulting in the use of 37 items in the survey. These 37 items were delivered to the survey firm on June 21, at which point they were transformed into an online format and underwent pre-testing. Following the incorporation of the pretest results, an actual survey was conducted in early July. After basic data coding and cleaning, the research team received the survey data on July 12.

A concise summary of the procedure is shown in Figure 1. After conducting the literature review, 55 survey items were developed. Through expert consultations, 37 items were selected from the pool. An online survey was conducted using the selected items. Exploratory factor analysis was employed for the surveyed items, resulting in the exclusion of one item that did not meet the minimum statistical criteria. Of the remaining 36 items, 23 were initially selected based on their relatively strong impact according to statistical criteria. Further considerations, including the balance between factors, theoretical relevance, and coverage of key aspects of drug use behavior, led to the final selection of 15 items. Confirmatory factor analysis (CFA) was performed on these 15 items to confirm their validity. The detailed analysis results are presented in the Results section.

The Korean Motivation Questionnaire developed in this study was based on Cooper’s [11] four-factor model of alcohol use modified for drug use. This model serves as a standard motivational framework. Consistent with the findings of Jones et al. [16], we integrated pain relief into a four-factor model. In addition, we incorporated Bennett and Holloway’s [19] findings on prescription drug misuse, items from the SUMM’s [12] eight subscales, and insights from Korean research conducted by Yu [21] and Yoo [22]. Following a complex process of scale development that involved translation, back-translation, expert consultation, and item vetting, the preliminary scale comprised 37 items with a 4-point Likert scale response format (ranging from 1 = “strongly disagree” to 4 = “strongly agree”). Table 1 lists the 37 preliminary items. The final version of the developed Korean items was used in the survey, and Table 1 presents a reiteration of the items translated into English. The English version of the items did not undergo rigorous academic processes, such as back-translation; rather, it was translated arbitrarily by the research team. The questions in the survey were numbered identically to those presented in the table and posed sequentially following the specified order.

### 2.3. Validity Verification Methods

To evaluate the instrument’s concurrent validity, we assessed personality traits previously identified as correlates of drug use motivation [17,18]. Complementarily, we included items based on the health belief model (HBM), which reflects the measure’s theoretical underpinnings premised on the rational anticipation of drug use repercussions [23,24,25,26]. The 15 items that we developed based on the HBM demonstrated a clear alignment. Focusing on drug use, five factors were measured (each comprising three questions): perceived sensitivity, perceived severity, perceived barriers, perceived benefits, and self-efficacy. Respondents rated the items on a 5-point Likert scale ranging from “strongly disagree” (1) to “strongly agree” (5). Appendix A provides a detailed list of the 15 HBM items.

To evaluate the instrument’s predictive validity, the participants were surveyed regarding their history of drug use and consumption of various drugs. They were presented with the following question: “Have you ever used any substances such as sedatives, marijuana, LSD, amphetamines, crack, cocaine, non-prescribed opioids, heroin, ecstasy, GHB, methadone, psychedelic mushrooms, or ketamine?” Participants chose either “yes” or “no”. The frequency of their responses was tallied to determine the extent of their drug experience.

To gain insight into the range of drug experiences, we classified commonly distributed drugs in Korea into five categories. Furthermore, the language was objective, neutral, and free of emotional and figurative expressions, and hedging was used to avoid bias. The five categories included drugs targeting weight loss, drugs designed for cognitive enhancement, traditional illicit drugs like marijuana and methamphetamine, substances known for their “sobering” effects that are frequently associated with entertainment venues, and newer drugs such as synthetic cannabinoids. Participants reported their drug exposure by indicating whether they had used drugs in each category via a “yes” or “no” response.

Considering the estimated drug use rate among Koreans is approximately 1%, it was anticipated that only a relatively limited number of participants out of the 250 surveyed in this study would report drug use experience. This poses a challenge when evaluating predictive validity. To overcome this limitation, we used proxy measures. Although alcohol is not classified as a drug, it is widely misused and was therefore chosen as a proxy substance. Using alcohol consumption, which is culturally prevalent despite the legal prohibition of drug use in Korea, as a surrogate variable has fundamental limitations. Nevertheless, despite these constraints, the decision was made to do so considering the existing research findings that demonstrate a substantial overlap and close association between motives for alcohol consumption and drug use at the individual level [12,13]. Therefore, the participants were asked about the frequency and amount of alcohol they consumed. The question “How often do you consume alcohol?” was used to assess the frequency. The response options ranged from “abstained for the past year” (1) to “consumed 4–5 times or more weekly” (7). For alcohol consumption, participants were asked, “How many standard drinks do you typically consume on a single occasion?”. Responses ranged from “one standard drink” (1) to “35 or more standard drinks” (17).

### 2.4. Data Analysis

To evaluate the validity of the instrument, we initially performed exploratory factor analysis (EFA) using the principal axis factor extraction method, covariance matrix, and varimax rotation. If the rotated factor loading, which indicates the direct influence of each factor on the respective variable, was 0.5, we considered the variable to belong to that factor. Subsequent to the EFA, we conducted CFA to outline the relationships between the latent and observed variables. The maximum likelihood estimation method was used. The model fit was evaluated using the chi square/degrees of freedom (CMIN/DF), comparative fit index (CFI), goodness-of-fit index (GFI), incremental fit index (IFI), and root mean square error of approximation (RMSEA). According to Hu and Bentler’s [27] model, a good fit is indicated by CFI, GFI, IFI, and TLI values surpassing 0.9. An RMSEA value ranging from 0.05 to 0.08 suggests a satisfactory model fit.

To evaluate internal consistency reliability, we computed Cronbach’s alpha. Reliability was further confirmed using factor analysis, average variance extracted (AVE), and intraclass reliability (ICR) values. To evaluate the criterion-related validity, we correlated the instrument with criterion measures while controlling for age, sex, education, income, and marital status. IBM SPSS Statistics 26 and IBM SPSS AMOS 21 were used for all the statistical analyses.

## 3. Results

### 3.1. Participants’ Characteristics

The demographic characteristics of the 250 participants are presented in Table 2.

### 3.2. Exploratory Factor Analysis

EFA was conducted on 37 items. The inverse covariance matrix was reviewed before the main analysis. A measure of sample adequacy (MSA) value below 0.6 was found for the item “Drug use is an act of defiance against authority”, resulting in its exclusion. After removing this item, the EFA yielded a Kaiser–Meyer–Olkin (KMO) value of 0.912. The results of Bartlett’s test of sphericity confirmed the suitability of the data for factor analysis, with a significance value of 5882.364 (*p* < 0.001). Additionally, all the remaining items exceeded the MSA value of 0.738. By applying Kaiser’s criterion, five factors with eigenvalues exceeding one emerged. The scree plot clearly identifies four to five factors.

After conducting multiple analyses to retain items with rotated factor loadings of at least 0.5 for one or more factors, a final set of 23 items was identified (Table 3). This refined item set presented a KMO metric of 0.929, and Bartlett’s test of sphericity yielded a significant value of 4601.168 (*p* < 0.001). All items exhibited MSA values higher than 0.623. According to Kaiser’s criterion, three factors with initial eigenvalues greater than one were identified. Five factors were identified, with rotated squared loadings exceeding 1, accounting for 66.70% of total variance. Their individual contributions were 28.759%, 12.256%, 8.98%, 8.78%, and 7.92%. Of these 23 items, the 15 that best represented the content of each factor were selected. Three items were selected for each factor to achieve a balanced distribution of items across factors.

After an examination of the item content of the five factors, they were labeled “enhancement motivation”, “coping motivation”, “social motivation”, “positive expectancy regarding drug effects”, and “negative expectancy regarding drug effects”. The factors of coping motivation and positive expectancy regarding drug effects initially presented more items than their counterparts. Certain adjustments were made to harmonize the number of items across factors. Specifically, within the coping motivation category, while items related to coping with stress and depression were retained, an additional item related to coping with physical pain was included, despite its slightly lower factor loading. For the positive expectations regarding the drug effects factor, three items describing arousal effects, fatigue recovery, and weight loss were selected because of their pronounced factor loadings.

The AVE values for Factors 1–5 were 0.507, 0.546, 0.495, 0.452, and 0.501, and the corresponding ICR values were 0.676, 0.720, 0.649, 0.601, and 0.665, respectively. The Cronbach’s alpha values for Factors 1–5 were 0.861, 0.863, 0.842, 0.795, and 0.773, respectively. Pearson’s correlation coefficients for the factors ranged from −0.239 to 0.633 (Table 4). Given that the square root of the AVE value for each factor exceeded its correlation with the other factors, it can be inferred that discriminant validity exists among the factors.

### 3.3. Confirmatory Factor Analysis

Following the EFA, CFA was conducted on 15 items. The aim was to assess whether the five-factor structure showed a better fit than a unidimensional model in which all 15 items were loaded on a single factor. The fit indices for the unidimensional model were CMIN/DF = 9.431 (*p* < 0.001), CFI = 0.649, IFI = 0.651, GFI = 0.526, TLI = 0.590, and RMSEA = 0.184, indicating an overall unsatisfactory fit. The five-factor structure showed improved fit indices, namely, CMIN/DF = 2.124 (*p* < 0.001), CFI = 0.958, GFI = 0.917, IFI = 0.959, TLI = 0.945, and RMSEA = 0.067, indicating a good fit. The indices of this model suggest a more accurate representation of the data than the unidimensional model. A visual representation of the CFA for the five-factor model is shown in Figure 2.

The distribution of scores for the five motives based on sex and age is presented in Table 5 below.

### 3.4. Criterion-Related Validity

To evaluate concurrent validity, the 15-item measure based on the HBM was correlated with five motivational factors for substance use (Table 6). Significant correlations were found among four of the five HBM dimensions in the expected directions. These results indicate that individuals who were motivated to use drugs were more likely to hold beliefs that made them susceptible to using drugs; believe that drugs are not dangerous; dismiss the idea that there are advantages associated with drug prevention education; and lack feelings of self-efficacy with regard to drug resistance.

Due to the limited number of participants who reported previous drug use, assessing predictive validity posed challenges. Among the participants, 7.6% (n = 19) reported using at least one drug: 68.42% (n = 13) had used a weight-loss drug, 4 had used a substance typically found in entertainment venues, and 2 had used a traditional illicit drug. To assess predictive validity, correlations were examined between prior drug use (yes or no) and the variety of substances used with the five factors. However, no significant correlations were observed.

In light of these challenges, an effort was made to indirectly assess predictive validity using the frequency and volume of alcohol intake as proxy variables. Compared with drug use, a higher prevalence of alcohol use was observed (n = 206, 82.4%). The mean frequency was 3.61, with a standard deviation of 1.86, whereas the mean quantity was 6.03, with a standard deviation of 3.79. Frequency of alcohol consumption showed a significant positive correlation with enhancement motivation (Factor 1) and coping motivation (Factor 2). Moreover, the quantity of alcohol consumed was significantly and positively correlated with enhancement motivation, coping motivation, social motivation, and positive expectations regarding the effects of drugs.

Although these observations are preliminary, they support the validity of the five motivational factors. However, the correlation between the frequency of drinking alcohol and amount of alcohol consumed was only suggestive of drug use. The sample size of drug users may not have been sufficiently large to obtain statistically significant results.

## 4. Discussion

In this study, five distinct motivational categories associated with substance use were identified: enhancement, coping, social, positive expectancy, and negative expectancy.

“Enhancement” motivation predominantly encompasses aspects such as enjoyment, sexual pleasure, and creativity. The inclusion of creativity as an enhancement motivation is notable considering its subtle difference from the more straightforward pleasure-seeking motives that are commonly linked to substance use. While creativity is often linked to outcomes, such as increased concentration in conventional contexts, its alignment with enhancement motivations has broader implications. This might suggest that users acknowledge certain qualities of substances such as LSD and marijuana that have been theorized to enhance creativity [28,29].

“Coping” motivation encompasses various aspects, including addressing boredom, stress, anxiety, depression, psychological distress, physical pain, negative memories, and the desire to escape reality. This distinctive classification contrasts with several studies such as Cooper’s [11] seminal work, Bennett and Holloway’s [19] examination of substance use motivations, and Drazdowski’s [20] meta-analysis, which portrayed substance use as a coping mechanism for depression, anxiety, physical pain, or boredom [30]. Interestingly, while the SUMM [12] intentionally separates feelings such as anxiety, depression, and boredom into distinct motivations, our findings suggest a blending of these aspects. This unique grouping may have resulted from cultural nuances [31], limitations in our survey design, or the possibility that addressing a wide range of substances prompted more comprehensive responses about their effects. Further in-depth research, including cross-cultural comparisons, is required to gain a better understanding of this phenomenon.

“Social” motivation comprises three aspects: mingling with others, participating in social life, and seeking acceptance within a group. Insights from contemporary motivational frameworks indicate a merging of what was previously categorized as social and conformity motives into a single social motive. This alignment may mirror the collectivism inherent in Korean culture [31]. In Korea, socialization often transcends individual interactions and involves integration into larger groups. Through this integration, individuals directly acknowledge and conform to the group’s inherent norms [32]. A study of Korean alcohol consumption behaviors suggested that alcohol is often consumed in a social context so that individuals can adhere to group norms [33]. This pattern implies potential similarities regarding drug use among Koreans.

“Positive expectancy” motivation comprises elements associated with enhancing concentration, arousal effects, fatigue recovery, and weight loss. Among these, the aspect related to improving concentration seemed less emphasized and was therefore excluded from the primary categorization. This difference aligns with the observed disparities in the perception and frequency of drug use between Western countries and South Korea. Recent studies in Western contexts indicate that pharmaceutical cognitive enhancers (PCEs) such as modafinil and methylphenidate are more commonly used to boost productivity [34]. In contrast, the Korean government expresses concerns about high-caffeine-beverage consumption [35]; however, these substances are not typically grouped with or perceived in a similar manner to psychotropic drugs. In the Korean context, drugs are less likely to be associated with improving concentration or functioning as sleep aids and are more likely to be linked with feelings of euphoria and loss of self-control. The diminished emphasis on concentration enhancement in Korean drug perception may have been influenced by cultural nuances. However, the study results could also have been influenced by the absence of participants with experience using “study aids” or PCEs.

“Negative expectancy” motivation encompasses sleep disorders, digestive disorders, and changes in appearance. Traditional motivational measurement tools typically focus on factors that facilitate or motivate specific behaviors, often overlooking inhibitory factors or potential deterrents [36]. However, given that our study population mainly consisted of members of the general population with limited drug use experience, it was considered crucial to include motives that discouraged drug use [37]. The prevailing notion, emphasized by prior research [38,39], is that the potential addictive properties of drugs are well ingrained within the participants’ awareness. Thus, when contemplating drug use, this inherent knowledge may not serve as an additional contemplative factor but rather a foundational principle. Owing to their unique hierarchical significance in the decision-making process, it seems plausible, following Conner and Armitage’s [40] health decision-making model, that concerns regarding addiction do not amalgamate with other negative expectancy motivations.

The motivations considered in this study showed significant correlations with the five constructs of the HBM: susceptibility, severity (threats), barriers, benefits (evaluations of drug prevention education), and self-efficacy [41]. However, it is worth noting that positive expectancy motivation correlated with only two of the HBM constructs. Similarly, beliefs about barriers were only correlated with negative expectations regarding drug effects. This singular relationship with barrier beliefs could be the result of an emphasis on external factors such as policies and education, which diverge from the typical intrinsic motivations associated with health behaviors [42]. The absence of a correlation between positive expectancy motivation and the other HBM factors poses a complicated issue for interpretation. One potential explanation is that individuals may overlook the potential societal consequences and implications of drug prevention when they have specific physiological expectations of drug use, such as arousal effects or fatigue recovery.

Because of the small sample size, detecting a significant correlation between drug use motives and actual drug use experiences was problematic. Therefore, alcohol consumption was used as the criterion variable. The results indicated that the motivation to use drugs was positively related to alcohol consumption, and enhancement and coping motivations were positively associated with the frequency of alcohol use. According to Ko and Sohn [43], regular alcohol consumption linked to workplace practices has a ritualistic nature in Korean culture. This could explain the findings related to the frequency of alcohol use. This suggests that the motive to use drugs for enhancement and coping may occur similarly in the context of drinking given the Korean workplace culture. Yang and Sohn [44] suggested that the quantity of alcohol consumed could reflect elements such as impulsivity and thrill seeking. This suggests that the motives to use drugs may be associated with certain personality characteristics consistent with those that may contribute to drinking patterns. Woicik et al.’s [26] study of substance use determinants partly supports this notion.

In addition to the limited number of drug users captured in the research data, demonstrating that the predictive power of the motivational measurement tool lies in the complexity of the social impact associated with drug use was challenging. For instance, there is no consensus on the specifics of the experimental substance use phase that precedes addiction. Petraitis, Flay, and Miller [45] conducted an in-depth analysis of substance use models, proposing that substance use behaviors are shaped by an intricate interplay of factors across three levels of influence—ultimate, distal, and proximal—encompassing social/interpersonal, cultural/attitudinal, and intrapersonal influences. Statistical analyses of the social factors associated with drug use have revealed inconsistencies, suggesting inherent diversity in consumption patterns [6,7]. Unraveling the root causes of substance use behavior becomes increasingly complex when cultural factors are considered. Much of the literature has adopted a Western-centric perspective, contributing to a lack of understanding of substance abuse in non-Western cultures. As revealed in this study, the intensity of social motives related to drug use among Koreans appears relatively low. To enhance the predictability of drug use behavior in future research, these cultural differences should be more intricately considered and incorporated.

The results of this study regarding the tool’s development align with established research on drug use motivations based on recognized motivational frameworks [11,12,16]. Similar to previous studies, this study identified three distinct primary motives: enhancement, coping, and social. The items included in both positive and negative expectancy motivations also aligned with the trends observed in earlier research. These results highlight that when studying drug use, it is essential to consider sociocultural nuances, especially in the Korean context [35]. These findings reflect the impact of foundational theories on the development of a questionnaire for assessing Korean drug use motives.

This study, primarily centered on tool development, provides inspiration for formulating drug intervention strategies in Korea, although it does not specifically assess drug use motives in the general population. When examining the distribution of the five drug use motives by sex and age, despite the limitations of a small sample size for an in-depth analysis, potential variations in motive levels were observed across the demographic groups. For instance, among males, there was a tendency for hedonistic motives to increase with age, whereas females in their 20s showed the highest hedonistic motives. Female participants in their 20s were also the only subgroup that exceeded three points in positive expectancies. Considering these findings, drug intervention policies tailored to different life stages are necessary, especially those emphasizing the lack of positive effects, such as weight loss, emotional well-being, and physical benefits, for Korean females in their 20s. It is essential to promote the awareness that drug use does not result in positive outcomes or alleviate emotional or physical difficulties.

Additionally, when examining the correlation between HBM items and alcohol consumption, the most significant negative expectations among the motives were found to be positively associated with perceived severity and perceived benefits but negatively associated with perceived barriers and unrelated to perceived susceptibility. This suggests that prevention strategies need to transcend emphasizing the harm caused by drugs and include mechanisms to prevent social stigmas, provide specific information about support services, and increase the accessibility of preventive actions. The observed positive correlation between hedonistic and coping motives and self-efficacy, as well as the static correlation with alcohol frequency and quantity, underscores the need to address drug use within the context of mental health rather than via merely legal deterrence or punishment. Consequently, this tool could be instrumental in implementing evidence-based drugs and health policies in South Korea.

### Limitations

This study has several limitations. First, items highlighted in previous studies as significant motives were not assigned to specific factors. These included items suggesting positive experiences that might fall under the expansion motive factor and those emphasizing self-medication using psychotropic drugs. A possible reason for this could be an oversight during the initial item selection phase, in which the representation of these motives may not have been adequately considered. Therefore, this area should be considered during instrument refinement.

Second, the predictive validity of this measure requires further investigation. Owing to the small number of participants who had used drugs, verifying predictive validity was problematic. Although alcohol consumption was adopted as a surrogate variable, there were inadequacies in the use of the surrogate itself considering the differences in legal status between drug use and alcohol consumption, among other factors. Future research should consider increasing the sample size to address these limitations.

Third, our study was tailored to the sociocultural nuances of Korea and included a wide range of narcotic and psychotropic drugs. However, substances such as psychotropic drugs, marijuana, and narcotics inherently differ in their motives for use and distribution dynamics. It is important to recognize the unique characteristics of each substance, and future studies should focus on developing a more disaggregated approach to improve the validity and reliability of the instrument.

Fourth, this study was based on self-reported data, so it is possible that the participants may have provided inaccurate information. Concerns about the potential social stigma associated with drug use may provoke individuals to underreport their experiences, creating a higher likelihood of social desirability bias in self-administered surveys on sensitive topics such as drug use [46]. Consequently, experiences of drug use and positive expectations may be underreported, whereas negative expectations may be overreported. Although self-administered surveys are appropriate for measuring motivation, caution must be exercised when considering the validity of these tools.

Finally, our study focused on individuals from the general population with no history of drug use or who were in the early stages of experimenting with drugs. It is important to note that attitudes and motivations toward narcotic and psychotropic drugs may differ significantly between users and non-users. Understanding these dynamics is critical for future research.

## 5. Conclusions

In conclusion, this study involved the development and validation of a tool designed to evaluate motivations for drug use among Koreans. Five distinct factors were identified: enhancement; coping; and social, positive, and negative expectancies. The tool designed is a concise, 15-item questionnaire, with each factor containing three items. Its streamlined design was intentionally chosen to enable the quick assessment of drug use motives. The simplicity of the questionnaire guaranteed wide accessibility, thereby facilitating and simplifying the collection of participant responses across a broad spectrum. Although certain aspects require refinement, this tool represents a groundbreaking initiative toward comprehending drug use motives in the Korean population. Furthermore, the tool holds substantial value in comprehending the psychological causes of drug use in this context. As Korea faces challenges posed by an increasing number of drug users, our tool is designed to facilitate evidence-based policy decisions and strategies. This will support the timely prevention and mitigation of emerging addiction issues.

## Figures and Tables

**Figure 1 healthcare-12-00086-f001:**
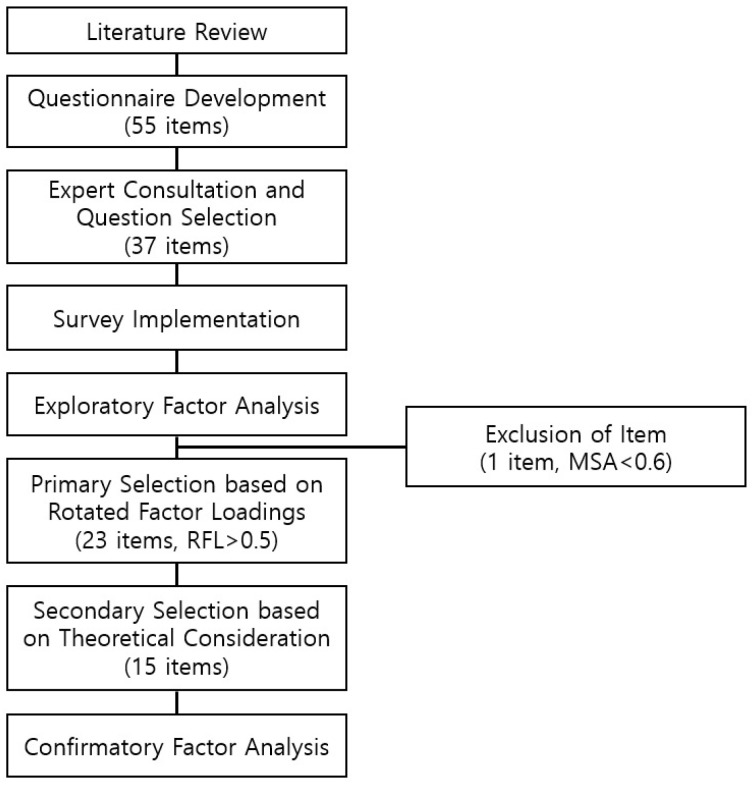
Tool development and factor analysis procedure.

**Figure 2 healthcare-12-00086-f002:**
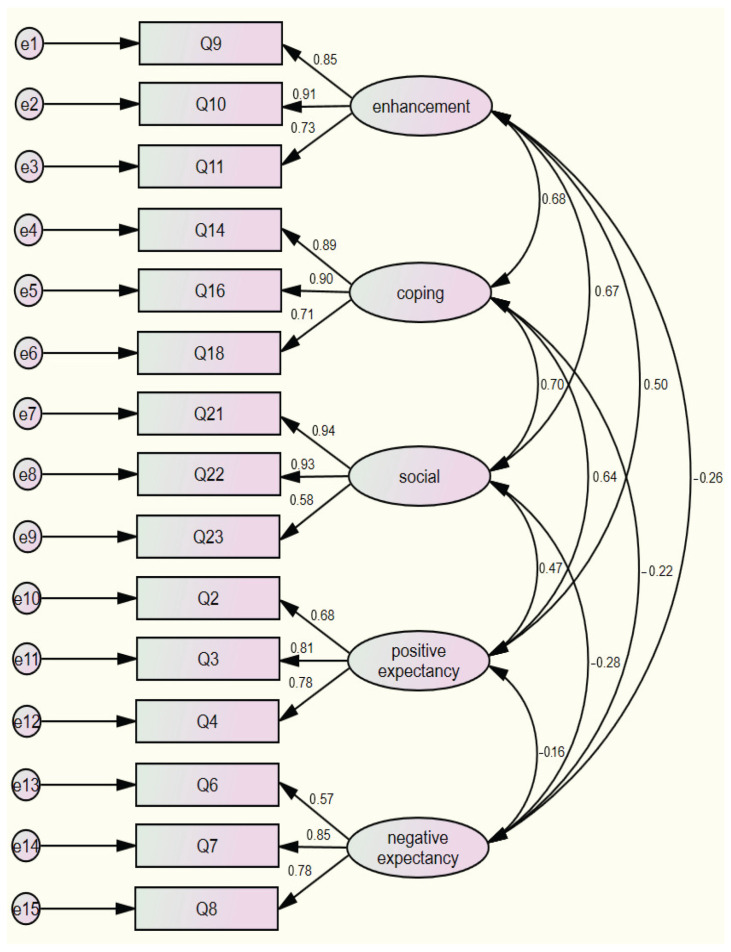
Confirmatory factor analysis for the five-factor model.

**Table 1 healthcare-12-00086-t001:** Preliminary items for the Korean Motivation Questionnaire.

Item	Mean	Standard Deviation	Skewness	Kurtosis
Q1. Using drugs helps improve memory	1.63	0.91	1.26	1.38
Q2. Using drugs helps with wakefulness	2.60	1.33	−0.29	−1.27
Q3. Using drugs helps with fatigue recovery	2.02	1.13	0.47	−0.87
Q4. Using drugs helps with weight loss due to increased physical energy	1.95	1.10	0.64	−0.58
Q5. Using drugs helps you socialize better	1.60	0.93	1.04	0.18
Q6. Using drugs causes sleep disorders	3.49	0.93	−1.29	1.82
Q7. Using drugs causes digestive problems	3.93	0.93	−0.49	−0.02
Q8. (Using drugs) It can change your appearance, such as causing you to lose teeth and acquire a distorted face	3.29	0.96	−1.13	1.16
Q9. Using drugs is a helpful way to have fun	1.64	0.88	1.09	0.02
Q10. Using drugs is a way to enhance sexual pleasure	1.77	0.93	0.87	−0.41
Q11. Using drugs is a way to enhance creativity	1.47	0.73	1.38	1.00
Q12. Using drugs is a way to escape from personal problems (stress, depression, etc.)	1.69	0.80	0.80	−0.43
Q13. Using drugs is a way to cope with boredom	1.55	0.74	1.13	0.36
Q14. Using drugs is a way to cope with stress	1.59	0.80	0.97	−0.46
Q15. Using drugs is a way to cope with anxiety	1.71	0.85	0.79	−0.61
Q16. Using drugs is a way to cope with depression	1.70	0.85	0.81	−0.57
Q17. Using drugs is a way to cope with mental pain	1.78	0.88	0.74	−0.58
Q18. Using drugs is a way to cope with physical pain	2.01	0.98	0.38	−1.15
Q19. Using drugs is a way to cope with bad past experiences	1.59	0.77	1.07	0.22
Q20. Using drugs is a way to escape from reality	1.95	1.05	0.60	−1.04
Q21. Using drugs is a way to fit in with other people	1.36	0.59	1.55	1.98
Q22. Using drugs is a way to enhance my social life	1.31	0.53	1.46	1.20
Q23. (Using drugs) is a way to be accepted into a social group	1.31	0.59	1.22	1.72
Q24. Increased tolerance can lead to addiction problems	3.95	0.73	−2.92	10.10
Q25. Using drugs shows personal weaknesses	2.84	1.01	−0.50	−0.83
Q26. Using drugs is a selfish behavior	3.32	0.80	−1.07	0.60
Q27. Using drugs is a way to harm yourself	3.74	0.58	−2.59	7.57
Q28. It is wrong to use drugs for any reason	3.45	0.79	−1.24	0.52
Q29. Using drugs has serious consequences for an individual	3.76	0.49	−2.17	5.30
Q30. Using drugs is a way to rebel against authority	2.38	1.04	0.23	−1.10
Q31. Using drugs can be a negative experience	3.63	0.65	−2.07	4.88
Q32. Using drugs can be a method of self-medication	1.50	0.80	1.47	1.23
Q33. Using drugs should be an individual’s choice	1.48	0.79	1.55	1.51
Q34. The use of drugs can occur in a normal society	1.62	0.85	1.24	0.72
Q35. Using drugs can be a positive experience	1.30	0.61	2.23	4.83
Q36. Using drugs can lead to addiction	3.76	0.61	−3.06	10.05
Q37. Using drugs is a victimless crime	1.64	1.00	1.39	0.56

**Table 2 healthcare-12-00086-t002:** Characteristics of participants.

Characteristic		N	%
Sex	Male	125	50.0
	Female	125	50.0
Age (years)	19–29	62	24.8
	30–39	62	24.8
	40–49	63	25.2
	50–59	63	25.2
Education	High school	43	17.2
	Junior college	44	17.6
	College	135	54.0
	Graduate school	28	11.2
Household income	Below KRW 3 million	63	25.2
	KRW 3.01–5 million	72	28.8
	KRW 5.01–7 million	57	22.8
	Above KRW 7.01 million	58	23.2
Marital status	Married/cohabiting	127	50.8
	Single/separated	123	49.2

**Table 3 healthcare-12-00086-t003:** Factors identified using exploratory factor analysis (N = 250).

Item	Adjusted Rotated Factor Loadings				
	Factor 1	Factor 2	Factor 3	Factor 4	Factor 5
Q1	0.046	0.206	0.136	**0.592**	−0.125
Q2	0.078	0.212	−0.058	**0.715**	0.091
Q3	0.105	0.287	0.185	**0.723**	−0.074
Q4	0.157	0.276	0.177	**0.666**	−0.073
Q5	0.039	0.175	0.493	**0.523**	−0.161
Q6	0.000	−0.037	−0.198	−0.077	**0.566**
Q7	−0.125	−0.135	0.045	−0.080	**0.800**
Q8	−0.058	−0.025	−0.059	0.011	**0.821**
Q9	**0.742**	0.321	0.235	0.093	−0.151
Q10	**0.809**	0.370	0.192	0.162	−0.070
Q11	**0.516**	0.455	0.280	0.283	−0.051
Q12	0.271	**0.738**	0.216	0.263	−0.083
Q13	0.253	**0.710**	0.359	0.169	−0.116
Q14	0.169	**0.806**	0.286	0.194	−0.092
Q15	0.154	**0.871**	0.174	0.239	−0.075
Q16	0.161	**0.845**	0.188	0.238	−0.072
Q17	0.157	**0.868**	0.138	0.220	−0.060
Q18	0.212	**0.652**	0.009	0.347	−0.030
Q19	0.193	**0.771**	0.304	0.204	−0.076
Q20	0.136	**0.663**	0.122	0.264	−0.021
Q21	0.311	0.476	**0.579**	0.144	−0.098
Q22	0.242	0.453	**0.652**	0.137	−0.177
Q23	0.188	0.234	**0.558**	0.145	−0.070

Factor loadings > 0.5 are presented in bold. Factor 1 = enhancement motivation, Factor 2 = coping motivation, Factor 3 = social motivation, Factor 4 = positive expectancies regarding drug effects, and Factor 5 = negative expectancies regarding drug effects.

**Table 4 healthcare-12-00086-t004:** Correlations among the factors.

Factor	F1	F2	F3	F4	F5
F1	(0.712)				
F2	0.633 **	(0.739)			
F3	0.629 **	0.596 **	(0.704)		
F4	0.418 **	0.550 **	0.378 **	(0.672)	
F5	−0.215 **	−0.170 **	−0.239 **	−0.103	(0.708)

The square roots of AVE values are shown in parentheses. F1: enhancement motivation, F2: coping motivation, F3: social motivation, F4: positive expectancies regarding drug effects, and F5: negative expectancies regarding drug effects. ** *p* < 0.01.

**Table 5 healthcare-12-00086-t005:** Average and standard deviation of five motives according to sex and age.

Sex	Age	Motives (Mean, SD)				
		Enhancement	Coping	Social	Positive Expectancy	Negative Expectancy
Male	19–29	1.46 (0.70)	1.90 (0.95)	1.26 (0.52)	2.49 (1.05)	3.96(0.74)
	30–39	1.56 (0.73)	1.96 (0.81)	1.30 (0.50)	2.57 (0.73)	4.09(0.83)
	40–49	1.71 (0.75)	1.84 (0.70)	1.40 (0.52)	2.62 (1.05)	3.91(0.67)
	50–59	1.73 (0.72)	1.55 (0.54)	1.27 (0.41)	2.04 (0.95)	3.79(0.90)
Female	19–29	1.76 (0.87)	1.98 (0.86)	1.39 (0.48)	3.03 (1.02)	4.28(0.68)
	30–39	1.60 (0.68)	1.63 (0.69)	1.40 (0.54)	2.36 (0.99)	4.33(0.61)
	40–49	1.57 (0.94)	1.69 (0.95)	1.32 (0.64)	2.32 (1.07)	4.13(0.80)
	50–59	1.67 (0.67)	1.64 (0.67)	1.31 (0.40)	2.50 (0.98)	3.81(0.87)
Total		1.63(0.75)	1.77 (0.78)	1.33 (0.50)	2.49 (1.00)	4.03 (0.78)

**Table 6 healthcare-12-00086-t006:** Correlations of related factors.

Dimension/Variable	Factor 1	Factor 2	Factor 3	Factor 4	Factor 5
Concurrent validity (HBM)					
Susceptibility	0.336 **	0.235 **	0.328 **	0.233 **	−0.108
Severity (dangers)	−0.145 *	−0.143 *	−0.280 **	−0.051	0.362 **
Barriers	−0.021	−0.022	−0.029	0.038	0.223 **
Benefits (advantages of drug prevention education)	−0.257 **	−0.232 **	−0.343 **	−0.114	0.417 **
Self-efficacy	−0.360 **	−0.267 **	−0.344 **	−0.171 *	0.213 **
Predictive validity					
Drug experience	0.062	−0.017	0.055	0.042	0.044
# of drugs used	−0.026	0.042	−0.062	0.052	−0.063
Drinking frequency	0.142 *	0.140 *	0.096	0.087	−0.130
Drinking volume	0.221 **	0.238 **	0.143 *	0.182 **	0.004

Factor 1: enhancement motivation, Factor 2: coping motivation, Factor 3: social motivation, Factor 4: positive expectancies regarding drug effects, and Factor 5: negative expectancies regarding drug effects. ** *p* < 0.01, * *p* < 0.05.

## Data Availability

Any queries regarding the data used in this study can be directed toward the corresponding author. The dataset used in this study is available upon request.

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
