# Peer review of "Development and Validation of a Questionnaire for Assessing Drug Use Motives in the General Population in South Korea"

_healthcare, 2023, doi:10.3390/healthcare12010086_

Round 1
Reviewer 1 Report
Comments and Suggestions for Authors
This manuscript describes the development of a multidimensional questionnaire to assess drug use motives in Korean context. The ms is detailed and describes different steps in the process, going from a pool of 55 literature-based items, to a pilot test (with 37 items and 23 items; Exploratory factor analysis; EFA), resulting in a final test (15 items; Confirmatory factor analysis; CFA). Although the process is described in detail, following the process is complicated because the 37-item version is represented in Appendix A, the 23-item version in Table 2 and the 15-item version in Figure 1. Some improvements appear to be possible:
Major remarks:
1. It is unclear how the reduction of the 37-item version to the 23-item version, and to the final 15-item version, was achieved. From the description (lines 237-254) it seems that the EFA was in effect done on a 36-item version (not the 37-item version) because item Q30 was excluded initially (line 239). The full results of this EFA are not reported. What are the criteria for excluding items from the 36-item version (factor loadings < 0.5?, MSA values?, or a combination of both?), and what are the criteria for excluding items from the 23-item version (face validity?). The results reported in lines 250-252 are from the 23-item version, but are the factor analysis results of the 15-item version at least as good as the result of the factor analysis of the 36-item version or the 23-item version? According to the literature, 3 items per factor should be considered an absolute minimum. It would be helpful if the results of this item-reduction process were represented in a more systematic way.
2. Apparently, a second questionnaire based on the HBM (lines 154-173) was developed by the authors. Because this questionnaire was not taken from the literature, it would be helpful to reproduce the questions in an Appendix, and to provide some statistical information on the factor structure and reliabilities.
3. With respect to the criterion-related validity, the authors report that only 19 participants reported drug use (line 189; this should be mentioned under Results, line 297!) and that, therefore, used alcohol was used as a proxy for drug use (line 198-204). These results should be mentioned under Results (line 297 sqq.; not under Methods), and the number of relevant answers should be mentioned (maybe in Table 4). Is the number of alcohol users substantially higher than the number of drug users?
4. Are the questionnaires used as presented to participants as reproduced in this manuscript? More explicitly, was the sequence of items identical to the numbering in appendix A or were the questions presented in randomized order? Are the questionnaires offered in English or Korean language? If the first is true, are the items as listed in the ms; if the latter is the case, some statement about translation into the English version is required somewhere in the ms.
Minor remark:
5. What is the rationale of using a 1-4 Likert scale for one questionnaire (line 149), a 1-5 scale for the other questionnaire (line 174) and a 1-7 scale for alcohol use (line 200)?
Author Response
Dear Reviewer
I would like to express my gratitude to all the reviewers who examined the manuscript. Based on the previous reviews, I was able to identify clear flaws in my manuscript and worked hard to write a better paper.
Reviewer 1
Comment 1
It is unclear how the reduction of the 37-item version to the 23-item version, and to the final 15-item version, was achieved. From the description (lines 237-254) it seems that the EFA was in effect done on a 36-item version (not the 37-item version) because item Q30 was excluded initially (line 239). The full results of this EFA are not reported. What are the criteria for excluding items from the 36-item version (factor loadings < 0.50), and what are the criteria for excluding items from the 23-item version (face validity?). The results reported in lines 250-252 are from the 23-item version, but are the factor analysis results of the 15-item version at least as good as the result of the factor analysis of the 36-item version or the 23-item version? According to the literature, 3 items per factor should be considered an absolute minimum. It would be helpful if the results of this item-reduction process were represented in a more systematic way.
Response: Thank you for your valuable comment. I agree that one of the major weaknesses of the early draft of this paper was the inconsistent presentation of the process of selecting survey items, which could potentially cause significant confusion for readers. While the research team attributed this to a lack of clarity in the narrative, they recognized that the more significant issue lies in the lack of intuitiveness and visibility, hindering readers’ understanding. To address this, Figure 1, detailing the tool development and factor analysis procedure, has been added to Section 2.2. Research process. In addition, the following description has been included:
A concise summary of the procedure is shown in Figure 1. After conducting the literature review, 55 survey items were developed. Through expert consultations, 37 items were selected from the pool. An online survey was conducted using the selected items. Exploratory factor analysis was employed for the surveyed items, resulting in the exclusion of one item that did not meet the minimum statistical criteria. Of the remaining 36 items, 23 were initially selected based on their relatively strong impact according to statistical criteria. Further considerations, including the balance between factors, theoretical relevance, and coverage of key aspects of drug use behavior, led to the final selection of 15 items. Confirmatory factor analysis (CFA) was performed on these 15 items to confirm their validity. The detailed analysis results are presented in the Results section.
Figure 1. Tool development and factor analysis procedure
Comment 2
Apparently, a second questionnaire based on the HBM (lines 154-173) was developed by the authors. Because this questionnaire was not taken from the literature, it would be helpful to reproduce the questions in an Appendix, and to provide some statistical information on the factor structure and reliabilities.
Response: Thank you for your comment. As mentioned, crucial information was missing from our readers. In Appendix B, we presented the mean, standard deviation, kurtosis, skewness, and Cronbach’s alpha values for the utilized Health Belief Model (HBM) items. In addition, we provided model fit indices for the constructed HBM items. To streamline the explanation and address the added complexity owing to the inclusion of the Appendix table, we have succinctly summarized it as follows:
The 15 items that we developed based on the HBM demonstrated a clear alignment. Focusing on drug use, five factors were measured (each comprising three questions): perceived sensitivity, perceived severity, perceived barriers, perceived benefits, and self-efficacy. Respondents rated the items on a 5-point Likert scale ranging from "strongly disagree" (1) to "strongly agree" (5). Appendix A provides a detailed list of the 15 HBM items.
Comment 3
With respect to the criterion-related validity, the authors report that only 19 participants reported drug use (line 189; this should be mentioned under Results, line 297!) and that, therefore, used alcohol was used as a proxy for drug use (line 198-204). These results should be mentioned under Results (line 297 sqq.; not under Methods), and the number of relevant answers should be mentioned (maybe in Table 4). Is the number of alcohol users substantially higher than the number of drug users?
Response: Thank you for your feedback, which has been validated and incorporated. I have moved the information on the frequency and percentage of drug and annual alcohol users, as described in the Methods section, to Section 3.4. Accordingly, I have made slight modifications to the descriptive approach. Additionally, I explicitly stated that there were more individuals with alcohol use experience than drug users (7.6% vs. 82.4%).
Due to the limited number of participants who reported drug use, assessing predictive validity poses challenges. Among the participants, 7.6% (n=19) reported using at least one drug: 68.42% (n=13) had used a weight-loss drug, four had used a substance typically found in entertainment venues, and two had used a traditional illicit drug. To assess predictive validity, correlations were examined between prior drug use (yes or no) and the variety of substances used with the five factors. However, no significant correlations were observed.
In light of these challenges, an effort was made to indirectly assess the predictive validity using the frequency and volume of alcohol intake as proxy variables. Compared with drug users, a higher prevalence of alcohol use was observed (n=206, 82.4%). The mean frequency was 3.61 with a standard deviation of 1.86, whereas the mean quantity was 6.03 with a standard deviation of 3.79. Frequency of alcohol consumption showed a significant positive correlation with enhancement motivation (Factor 1) and coping motivation (Factor 2). Moreover, the quantity of alcohol consumed was significantly and positively correlated with enhancement motivation, coping motivation, social motivation, and positive expectations regarding the effects of drugs.
Comment 4
Are the questionnaires used as presented to participants as reproduced in this manuscript? More explicitly, was the sequence of items identical to the numbering in appendix A or were the questions presented in randomized order? Are the questionnaires offered in English or Korean language? If the first is true, are the items as listed in the ms; if the latter is the case, some statement about translation into the English version is required somewhere in the ms.
Response: Thank you for your insightful comments. The survey was conducted in Korean, and a separate English version was not developed. The item numbers correspond to the numbers assigned to the survey. Acknowledging this point, the following description has been added to Section 2.2 in the paper:
The survey used the final version of the developed Korean items, and Table 1 presents a reiteration of the items translated into English. The English version of the items did not undergo rigorous academic processes, such as back-translation; rather, it was translated arbitrarily by the research team. The questions in the survey were numbered identically to those presented in the table and posed sequentially following the specified order.
Comment 5
What is the rationale of using a 1-4 Likert scale for one questionnaire (line 149), a 1-5 scale for the other questionnaire (line 174) and a 1-7 scale for alcohol use (line 200)?
Response: Thank you for your insightful comments. The variation in the measurement categories for each item was due to the reliance on different key references during the development process of each item. This decision was made to enhance future comparability of the developed tool. To measure alcohol frequency and quantity, we adopted the methodology of the Korean National Health and Nutrition Examination Survey, considering the uniqueness of Korean drinking patterns. Therefore, the categorization may differ somewhat from conventional approaches. While we omitted such an explanation, as the differences in measurement categories for each item are not critical for the statistical validation of this study, if the reviewer deems this information important, we will be sure to include the relevant details in the manuscript.
Tip: Concise writing means using the fewest words necessary. One way to achieve conciseness is to use a direct verb rather than its noun form.
For example,
Original: We performed an evaluation of all methods used previously.
Revised: We evaluated all methods used previously.
Reviewer 2 Report
Comments and Suggestions for Authors
The paper by Yang and colleagues describes an interesting topic in analyzing predictory factors of drug abuse. It is generally well written and clear. However, it lacks of the most important information: how can this tool effectively improve the situation in this field? May the number of people affected by dependency be reduced?
Some of the predictory factors cited appear to be obvious and evident.
Author Response
Dear Reviewer
I would like to express my gratitude to all the reviewers who examined the manuscript. Based on the previous reviews, I was able to identify clear flaws in my manuscript and worked hard to write a better paper.
Comment 1
It lacks of the most important information: how can this tool effectively improve the situation in this field? May the number of people affected by dependency be reduced? Some of the predictory factors cited appear to be obvious and evident.
Response: Thank you for your comment. The points you raised are crucial, and they have unfortunately been omitted from the initial draft. To enrich the recommendations for intervention strategies, I have added Table 4 to Section 3.3, including the average values of the five motives based on gender and age. The analysis revealed potential differences in drug use motives based on sex and age, and such information is expected to contribute to future evidence-based policy development. Based on this study, we have added the following two paragraphs to the Discussion section. We appreciate your feedback as it has undoubtedly enhanced the overall completeness of this paper.
This study, primarily aimed at tool development, provides inspiration for formulating drug intervention strategies in Korea, although it did not specifically assess drug use motives in the general population. When examining the distribution of the five drug use motives by sex and age, despite the limitations of a small sample size for an in-depth analysis, potential variations in motive levels were observed across the demographic groups. For instance, among males, there was a tendency for hedonistic motives to increase with age, whereas females in their 20s showed the highest hedonistic motives. Female participants in their 20s were also the only subgroup that exceeded three points in positive expectancies. Considering these findings, drug intervention policies tailored to different life stages are necessary, especially those emphasizing the lack of positive effects, such as weight loss, emotional well-being, and physical benefits, among Korean females in their 20s. It is essential to promote the awareness that drug use does not result in positive outcomes or alleviate emotional or physical difficulties.
Additionally, when examining the correlation between HBM items and alcohol consumption, the most significant negative expectations among motives were found to be positively associated with perceived severity and perceived benefits but negatively associated with perceived barriers and unrelated to perceived susceptibility. This suggests that prevention strategies need to go beyond emphasizing the harm caused by drugs and include mechanisms to prevent social stigma, provide specific information about support services, and increase the accessibility of preventive actions. The observed positive correlation between hedonistic and coping motives and self-efficacy, as well as the static correlation with alcohol frequency and quantity, underscores the need to address drug use within the context of mental health rather than merely legal deterrence or punishment. Consequently, this tool could be instrumental in implementing evidence-based drugs and health policies in South Korea.
Reviewer 3 Report
Comments and Suggestions for Authors
The Materials and Methods section outlines the participant selection process, the development of the questionnaire, the measures used, data analysis methods, and validation procedures. Could you provide more information on the demographics of the selected participants, such as their age, gender distribution, and geographic locations within Korea, to better understand the representativeness of the sample? Can you provide insights into how the identification of these five distinct motivational categories can inform potential prevention and intervention strategies in Korea, considering the unique sociocultural nuances and the varying perceptions of different substances among the population?
Comments on the Quality of English LanguageOverall the English writing is satisfying.
Author Response
Dear Reviewer
I would like to express my gratitude to all the reviewers who examined the manuscript. Based on the previous reviews, I was able to identify clear flaws in my manuscript and worked hard to write a better paper.
Comment 1
Could you provide more information on the demographics of the selected participants, such as their age, gender distribution, and geographic locations within Korea, to better understand the representativeness of the sample?
Response: Thank you for your valuable comment. I would like to provide information on the observed differences between the distributions of our study participants and the general population.
As of June 2023, according to the population data released by Statistics Korea, the gender ratio in South Korea is 99.3, with slightly more females than males. The age distribution is as follows: 10s, 9.1%; 20s, 12.2%; 30s, 12.8%; 40s, 15.5%; 50s, 16.8%; 60s, 14.7%; and 70s or older, 12.1%. In our study, the 20–30s age group was slightly oversampled.
Based on education level data provided by the Ministry of Education in 2021, for the population aged 25–64 years, 11% graduated from middle school, 39% graduated from high school, and 50% had a university degree or higher. The participants in our study exhibited a relatively higher educational level than the general population, suggesting oversampling of younger age groups.
According to data from Statistics Korea in 2023, the value of 1 million won is approximately $750. The average real living cost for an individual living alone in South Korea is reported to be 2.2 million won, with a median value of 2 million won. Additionally, the average monthly income of an individual worker is approximately 3.1 million won, whereas a four-person household earns an average monthly income of 6.6 million won. Although the limitations of our measurement method prevent a definitive conclusion from being drawn, it appears that the household income levels of our study participants did not significantly differ from those of the general Korean population.
Marital status among Koreans varies depending on the survey, but according to data such as the Future Household Prediction by Statistics Korea, as of 2017, marriage and cohabitation rates were approximately 53%. Considering that our study participants were relatively young, the marital rate seems to reflect the situation of the general Korean population.
In summary, the survey participants in our study have a characteristic of being relatively younger, but they seem to well represent the overall population. If you consider this information to be crucial for the readers, we will include it in the manuscript.
Comment 2
Can you provide insights into how the identification of these five distinct motivational categories can inform potential prevention and intervention strategies in Korea, considering the unique sociocultural nuances and the varying perceptions of different substances among the population?
Response: Thank you for your insightful comment. I consider the points you raised crucial, and they have unfortunately been omitted from the initial draft. To enrich the recommendations for intervention strategies, I have added Table 4 to Section 3.3, including the average values of the five motives based on sex and age. The analysis revealed potential differences in drug use motives based on sex and age, and such information is expected to contribute to future evidence-based policy development. Based on this study, we have added the following two paragraphs to the Discussion section. We appreciate your feedback as it has undoubtedly enhanced the overall completeness of this paper.
This study, primarily aimed at tool development, provides inspiration for formulating drug intervention strategies in Korea, although it did not specifically assess drug use motives in the general population. When examining the distribution of the five drug use motives by sex and age, despite the limitations of a small sample size for an in-depth analysis, potential variations in motive levels were observed across the demographic groups. For instance, among males, there was a tendency for hedonistic motives to increase with age, whereas females in their 20s showed the highest hedonistic motives. Female participants in their 20s were also the only subgroup that exceeded three points in positive expectancies. Considering these findings, drug intervention policies tailored to different life stages are necessary, especially those emphasizing the lack of positive effects, such as weight loss, emotional well-being, and physical benefits, among Korean females in their 20s. It is essential to promote the awareness that drug use does not result in positive outcomes or alleviate emotional or physical difficulties.
Additionally, when examining the correlation between HBM items and alcohol consumption, the most significant negative expectations among motives were found to be positively associated with perceived severity and perceived benefits but negatively associated with perceived barriers and unrelated to perceived susceptibility. This suggests that prevention strategies need to go beyond emphasizing the harm caused by drugs and include mechanisms to prevent social stigma, provide specific information about support services, and increase the accessibility of preventive actions. The observed positive correlation between hedonistic and coping motives and self-efficacy, as well as the static correlation with alcohol frequency and quantity, underscores the need to address drug use within the context of mental health rather than merely legal deterrence or punishment. Consequently, this tool could be instrumental in implementing evidence-based drugs and health policies in South Korea.
Reviewer 4 Report
Comments and Suggestions for Authors
Dear Authors,
I have read the manuscript and I send you my comments:
1) Introduction is very long, please reduce it
2) Methods: please clarify the experimental protocolo it is very hard to read
3) results please add the data and the differentiation between patinets, age, sex, job and drug use
4) Results are very hard to evaluate please add tables or figures
5) Discussion: I have not understand it, please write again
6) Table 1: please change gender with sex
Author Response
Dear Reviewer
I would like to express my gratitude to all the reviewers who examined the manuscript. Based on the previous reviews, I was able to identify clear flaws in my manuscript and worked hard to write a better paper.
Comment 1
Introduction is very long, please reduce it
Response: Thank you for this insightful suggestion. We agree that the Introduction may be somewhat lengthy and verbose. Consequently, two significant modifications have been made. First, I have relocated the paragraph explaining ESU entirely to the Discussion section, illustrating the challenges of predicting drug use through surveys conducted on the general population. Second, I have removed the paragraph listing studies based on the DMM framework, providing only brief references in the preceding paragraph. This has resulted in a more concise and visually accessible introduction.
Comment 2
Methods: please clarify the experimental protocol. it is very hard to read
Response: Thank you for your comment. I agree that one of the major weaknesses of the early draft of this paper was the inconsistent presentation of the process of selecting survey items, which could potentially cause significant confusion for readers. While the research team attributed this to a lack of clarity in the narrative, they recognized that the more significant issue lies in the lack of intuitiveness and visibility, hindering readers' understanding. To address this, Figure 1, detailing the tool development and factor analysis procedure, has been added to Section 2.2. In addition, the following description has been included:
A concise summary of the procedure is shown in Figure 1. After conducting the literature review, 55 survey items were developed. Through expert consultations, 37 items were selected from the pool. An online survey was conducted using the selected items. Exploratory factor analysis was employed for the surveyed items, resulting in the exclusion of one item that did not meet the minimum statistical criteria. Of the remaining 36 items, 23 were initially selected based on their relatively strong impact according to statistical criteria. Further considerations, including the balance between factors, theoretical relevance, and coverage of key aspects of drug use behavior, led to the final selection of 15 items. Confirmatory factor analysis (CFA) was performed on these 15 items to confirm their validity. The detailed analysis results are presented in the Results section.
Comment 3
results please add the data and the differentiation between patients, age, sex, job and drug use
Response: Given that the number of participants reporting drug use experiences was extremely small–approximately three individuals out of the total 250 participants–reporting the distribution of drug use across groups was not deemed significant. This study focused on developing a tool to understand motivations for drug use among the general population, not among patients, reflecting the ambiguous expectations that individuals who have not yet encountered drugs or are not in a severe state of addiction might have. Therefore, in this study, the reporting of drug use, including group-level comparisons, was omitted. In future large-scale surveys using this tool, efforts should be made to provide this information. We appreciate the reviewer’s understanding.
However, upon reviewing your feedback, I discovered that the distribution of the five motivation factors based on sociodemographic characteristics was not included in the initial draft. As a result, I have incorporated this information into Table 4. Once again, I appreciate your valuable insight.
Comment 4
Results are very hard to evaluate. please add tables or figures
Response: Thank you for your feedback on our manuscript. We appreciate your attention to the Results section and your suggestion to add tables and figures.
Based on your comments, we have carefully assessed the current state of the manuscript. We believe that the inclusion of tables and figures follows standard practice and effectively supports our results. While we acknowledge the importance of clear visualization, we feel confident in the ability of existing graphical elements to accurately convey the findings.
We want to ensure that we have taken your input seriously and made efforts to enhance the overall clarity of the manuscript without introducing significant modifications to the tables and figures. We believe that this paper maintains a balance between clarity and conciseness. If you have any specific recommendations or additional concerns, we are open to further discussions and adjustments to address your expectations.
Thank you once again for your time and valuable insights
Comment 5
Discussion: I have not understand it, please write again
Response: I am not sure what you mean. I have revised most of the paragraphs in the Discussion section to improve the sentence structures. I have also brought some paragraphs from the introduction that were suitable for the discussion and modified them to fit the context. If there are still areas that you find problematic, please specifically point them out and make further adjustments.
Comment 6
Table 1: please change gender with sex
Response: I have made the necessary revisions as per your suggestion.
Reviewer 5 Report
Comments and Suggestions for Authors
Thank you for inviting me to review this manuscript, in the attached document you will find the revision.
Kind regards.

Author Response
Dear Reviewer
I would like to express my gratitude to all the reviewers who examined the manuscript. Based on the previous reviews, I was able to identify clear flaws in my manuscript and worked hard to write a better paper.
Comment 1
Since your population was healthy, please indicate this in the title.
Response: Thank you for your comment. I have considered your opinion and revised the title as follows:
Development and Validation of a Questionnaire for Assessing Drug Use Motives in the General Population of South Korea
Comment 2
In the methodology they state that the sample selection was a representative population, however, I do not understand the result obtained afterwards with such a low percentage of consumer experience. Could you please clarify this?
Response: I acknowledge some uncertainty regarding the precise interpretation of your inquiry, but I would like to provide an explanation to the best of my abilities. As mentioned in the Introduction, the prevalence of drug use in Korea, including both legal and illegal substances, is approximately 1.0%. In our study, which surveyed 250 participants, excluding those who reported using unprescribed weight loss medications (such as Saxenda and Xenical-Alli), the number of individuals reporting drug use amounted to three, representing 1.2% of the study participants. This aligns closely with the estimates provided by Korean law enforcement authorities. Therefore, there were no significant issues regarding the representativeness of our survey.
If your inquiry pertains to the reasons why drug use in Korea appears to be lower than that in other countries, further exploration may be necessary. One possible contributing factor is the stringent anti-drug policies implemented by military regimes that governed Korea for an extended period, possibly as a means of securing legitimacy. I hope that this provides a satisfactory response to your question, and if you find it necessary, I am willing to incorporate such explanations into the manuscript.
Comment 3
I understand that you have relied on alcohol consumption and questionnaires for alcohol consumption, it gives a very interesting support to your research. However, have you taken into account that alcohol consumption is not illegal, but drug consumption is?
Response: Thank you for your insightful comment. The research team may not have sufficiently considered aspects that were already in their awareness, and there may not have been adequate reflection on these points in the initial draft. In response, the following sentence has been inserted into section 2.3. In addition, the corresponding aspects have been emphasized in section 4.1.
Using alcohol consumption, which is culturally prevalent despite the legal prohibition of drug use in Korea, as a surrogate variable has fundamental limitations. Nevertheless, despite these constraints, the decision was made to do so considering the existing research findings that demonstrate a substantial overlap and close association between motives for alcohol consumption and drug use at the individual level [12-13].
Comment 4
In lines 226-229 he describes the characteristics of the sample, in which he describes the formation of the sample and then in table 1 he again presents the same data. Please do not duplicate the information, as it does not provide additional information. Please reconsider the wording of these lines.
Response: Thank you for your insightful suggestion. I have incorporated your suggestion to avoid redundancy by removing unnecessary descriptions in section 3.1, considering that the participants’ information is already presented in Table 1.
Comment 5
The references used in this manuscript are appropriate and no unnecessary self-citations have been detected. Congratulations on the use of up-to-date references, most of which are less than 5 years old. Please review those older than 2010 and try to provide more current references if possible. Revise the following references for formatting errors: 1, 2, 5, 26, 27, 29, 33, 34, 35, 40, 42.
Response: I always strive to adhere to the formatting requirements of the Healthcare journal in terms of the references. Thank you for pointing out the errors during the citation process. Owing to the deadline for submitting revisions and the current situation of the research team, there may be some insufficient areas as perceived by the reviewer. I will incorporate feedback from the editorial team in the subsequent editing process to make further improvements.
Reviewer 6 Report
Comments and Suggestions for Authors
The topic of this manuscript is interesting, but I think the work can become more accessible and readable by giving it a clearer structure.
In fact, the title gives a clue: development and validation. Also, the steps followed are clearly described in the Abstract. In the main text however, the presentation is a long story in which I lost track. This counts for the Methods section as well as the Results and Discussion sections.
I suggest to add a flow diagram, showing the subsequent steps. Forthermore, follow these steps in the main text and use appropriate (sub)headings. For instance, headings as used now for 2.2 and 2.3 are not clear. You may use comparable headings for the various sections to create uniformity and to keep the reader on track.
Furthermore, the Introduction is quite lengthy and contains information that may better fit in the Discussion, especially the details given in the second part.
In addition to this, I have some other recommendations.
Adapt the title. Suggestion: Development and validation of a questionnaire to investigate motives for drug use in South Korea (or: by South Korean adults).
The keywords may contain additional terms, like narcotics, psychotropic drugs, cannabis. (See lines 118, 119 where the focus is clearly given.)
Line 82: DMM, please add from which source this report (?) is. Same for SUMM in line 90. Where do these come from, what is their status?
Section 2.2 is not clear to me. What is ‘a month-long review of the literature beginning in May 2023’? Did the review take one month and was it done in May? I assume that the literature search covered more than just one month.
Line 150: Appendix A. I suggest to include this information as a table in the main text, as it is crucial to follow the story. It is not additional material.
Then, from 37 items 23 were selected. Which items? Table 2 gives Q1-Q23. How do they correspond to the list in Appendix A?
Section 2.3 is very confusing in structure. See my earlier comment on this. It also contains results and should be transferred to the appropriate section (for instance, but not only, line 188-195, 203-204) and even discussion.
The subjects answered questions. Were they honest, you believe? There may be a bias, shame, in telling about drug use.
Section 3.2. Refer to the list in Appendix A (and include that list in the main text).
On several places in the manuscript the authors use .xxx for presenting numbers or figures. Replace by 0.xxx.
What are ‘rotating factor loadings’ (line 245), and compare with Table 2 where it reads ‘adjusted rotated factor loading’? Not found in the methods section.
Provide a clearer legend to Table 2. The legend should explain what is presented in the table.
At the end there is an acknowledgement in Korean language.
Especially the structure of the manuscript and thus the way in which the work is presented to the reader should be improved. Make a roadmap (figure) and follow this flow. Then also consider whether all information is on the right place and adapt where appropriate.
Author Response
Dear Reviewer
I would like to express my gratitude to all the reviewers who examined the manuscript. Based on the previous reviews, I was able to identify clear flaws in my manuscript and worked hard to write a better paper.
Comment 1, 2, 8, 9, 11
- I suggest to add a flow diagram, showing the subsequent steps. Furthermore, follow these steps in the main text and use appropriate (sub)headings. For instance, headings as used now for 2.2 and 2.3 are not clear. You may use comparable headings for the various sections to create uniformity and to keep the reader on track.
- Then, from 37 items 23 were selected. Which items? Table 2 gives Q1-Q23. How do they correspond to the list in Appendix A?
- Section 2.3 is very confusing in structure. See my earlier comment on this. It also contains results and should be transferred to the appropriate section (for instance, but not only, line 188-195, 203-204) and even discussion.
- Section 3.2. Refer to the list in Appendix A (and include that list in the main text).
Response: Thank you for your insightful comment. The review comments point towards a consistent issue, and I would like to address them collectively. The initial draft of this paper failed to present the research procedures clearly and concisely, causing significant confusion for the readers. In response to your feedback, I have made comprehensive revisions to enhance readability and address the highlighted issues.
First, I have clarified the meaning by renaming sections 2.2. and 2.3. to “Survey Item Development and Selection Process” and “Validity Verification Methods,” respectively. In section 2.2. I have inserted the flowchart you mentioned to provide a visual representation of the research flow, allowing readers to understand the process at a glance.
In each section, I have attempted to move the relevant parts corresponding to the analysis results to the “Results” section as much as possible. However, I have retained only the essential information that readers should know before delving into the analytical results. Additionally, I inserted Table 1, originally included in Appendix A, into the main body of section 2.2. This process has made the flow of the paper smoother than the draft, and the research team is satisfied with the outcomes. We appreciate the reviewer’s feedback.
Comment 2
the Introduction is quite lengthy and contains information that may better fit in the Discussion, especially the details given in the second part.
Response: Thank you for this insightful suggestion. We agree that the Introduction may be somewhat lengthy and verbose. Consequently, two significant modifications have been made. First, I have relocated the paragraph explaining ESU entirely to the Discussion section, illustrating the challenges of predicting drug use through surveys conducted on the general population. Second, I have removed the paragraph listing studies based on the DMM framework, providing only brief references in the preceding paragraph. This resulted in a more concise and visually accessible introduction.
Comment 3
Adapt the title. Suggestion: Development and validation of a questionnaire to investigate motives for drug use in South Korea (or: by South Korean adults)
Response: Thank you for your valuable comment. I have made an effort to incorporate your insights into the title of the manuscript as much as possible. Additionally, considering the suggestion from another reviewer that the study’s focus on a healthy population should be explicitly reflected in the title, I have taken both opinions into account and revised the title as follows:
Development and Validation of a Questionnaire for Assessing Drug Use Motives in the General Population of South Korea
Comment 4
The keywords may contain additional terms, like narcotics, psychotropic drugs, cannabis. (See lines 118, 119 where the focus is clearly given.)
Response: Thank you for your comment. I have added the keywords narcotics, psychotropic drugs, and cannabis, excluding validity and reliability. I anticipate that these keywords will help readers to quickly grasp and search for information on how narcotics are classified and addressed in South Korea.
Comment 5
Line 82: DMM, please add from which source this report (?) is. Same for SUMM in line 90. Where do these come from, what is their status?
Response: We appreciate this feedback and realize that our writing may have caused confusion. For the DMM and SUMM tools that you pointed out, the respective sources of these tools are mentioned at the end of the sentences (References 12 and 13). This level of description allows readers to verify the original sources. If your inquiry is different, please let me know, and I will make immediate adjustments.
Comment 6
Section 2.2 is not clear to me. What is ‘a month-long review of the literature beginning in May 2023’? Did the review take one month and was it done in May? I assume that the literature search covered more than just one month.
Response: Thank you for pointing this out. We appreciate this insightful feedback. As you have pointed out, the previous description of the research process is overly general. In response, we have enhanced the investigation process as follows:
To develop the questionnaire, the research team initiated the study in early April 2023 through a comprehensive literature review. In May, references relevant to the identified theories were gathered to inform the development of the research tools. By the end of May, a draft of the survey instrument comprising 55 items related to motivations for drug use was created. In early June, expert translation and back-translation were performed on these items, and detailed consultations were held regarding the content of the questions. The expert committee consulted for this research comprised professionals from diverse fields including healthcare, pharmacy, and various disciplines in sociology and Korean linguistics. During this process, 18 items were deemed inappropriate and were excluded, resulting in the use of 37 items in the survey. These 37 items were delivered to the survey firm on June 21, when they were transformed into an online format and underwent pre-testing. Following the incorporation of the pretest results, an actual survey was conducted in early July. After basic data coding and cleaning, the research team received the survey data on July 12.
Comment 7
Line 150: Appendix A. I suggest to include this information as a table in the main text, as it is crucial to follow the story. It is not additional material.
Response: Thank you for the comment. Owing to the extensive content in the table, I had inserted it into the Appendix to avoid disrupting the flow for the reader. However, considering the significance of the table to the core content of the study, I have decided, as you suggested, to incorporate it into the main body of the paper.
Comment 10
The subjects answered questions. Were they honest, you believe? There may be a bias, shame, in telling about drug use.
Response: Social desirability bias is a valid and expected concern in this study, given the sensitive nature of the topic, particularly in the context of self-reported drug use data. Although this limitation was absent in the initial draft, I addressed this concern by incorporating the following statement. Once again, I appreciate your thoughtful suggestions.
Fourth, this study was based on self-reported data, which raises the possibility that participants may have provided inaccurate information. Concerns about the potential social stigma associated with drug use may lead individuals to underreport their experiences, creating a higher likelihood of social desirability bias in self-administered surveys on sensitive topics such as drug use [50]. Consequently, experiences of drug use and positive expectations may be underreported, whereas negative expectations may be overreported. Although self-administered surveys are appropriate for measuring motivation, caution must be exercised when considering the validity of these tools.
Comment 12
On several places in the manuscript the authors use .xxx for presenting numbers or figures. Replace by 0.xxx.
Response: Thank you for pointing this out. I addressed this issue using the find function in the Word program to locate and correct instances where there were missing leading zeros before decimal points.
Comment 13
What are ‘rotating factor loadings’ (line 245), and compare with Table 2 where it reads ‘adjusted rotated factor loading’? Not found in the methods section.
Response: While a detailed explanation of the adjusted rotated factor loading was not explicitly provided, it is stated in line 187 that exploratory factor analysis was conducted using the varimax technique for factor rotation. Recognizing the potential lack of clarity in the terminology, the following sentence has been added to the Data Analysis section.
If the rotated factor loading, which indicates the direct influence of each factor on the respective variable, was 0.5, we considered the variable to belong to that factor.
Comment 14
Provide a clearer legend to Table 2. The legend should explain what is presented in the table.
Response: We agree with the feedback that the legend was not clearly written, which made it difficult to understand the table. To enhance reader comprehension, we have revised the table categories to align them with the legend terms. If you believe that additional information is necessary beyond what has been provided, please let me know, and I will promptly incorporate any required changes.
Comment 15
At the end there is an acknowledgement in Korean language.
Response: Thank you for pointing this out. I encountered some confusion while writing the acknowledgement as I did not receive a specific format from the funding organization. However, I have now included it as follows by specifying the funding institution:
This research was supported by a grant from the Ministry of Food and Drug Safety in 2023.
Tip: A dummy subject generally serves as a grammatical filler. These sentences, which generally begin with “It” or “There,” should be rephrased to achieve tighter, concise writing.
For example,
Original: There were many people standing in line.
Revised: Many people were standing in line.
Round 2
Reviewer 5 Report
Comments and Suggestions for Authors
Dear authors,
Thank you for taking my comments into account. I note that you have kept the references prior to 2010, are there no more current references on the subject, and are these older references really relevant to the present work? I ask because they are numerous.
Kind regards
Author Response
Thanks for your valuable comment. I removed 5 outdated or unnecessary references. My article was proofread by a professional English proofreader.

Reviewer 6 Report
Comments and Suggestions for Authors
The manuscript has definitely improved from the revision. The comments and suggestions have been well elaborated. In my opinion, the manuscript is now acceptable for publication in Healthcare.
Comments on the Quality of English LanguageMaybe a few linguistic issues to be corrected, but that wil come during the production phase. I am not a native English speaker either.
Author Response
Thanks for your comment. My article was proofread by a professional English proofreader.
